# Comparing the Effects of Bafa Wubu Tai Chi and Traditional He-Style Tai Chi Exercises on Physical Health Risk Factors in Overweight Male College Students: A Randomized Controlled Trial

**DOI:** 10.3390/ijerph20146323

**Published:** 2023-07-08

**Authors:** Yantao Niu, Rojapon Buranarugsa, Piyathida Kuhirunyaratn

**Affiliations:** 1Exercise and Sport Sciences Program, Graduate School, Khon Kaen University, Khon Kaen 40002, Thailand; yantao.ni@kkumail.com; 2Faculty of Physical Education, Jiaozuo Normal College, Jiaozuo 454000, China; 3Physical Education Program, Faculty of Education, Khon Kaen University, Khon Kaen 40002, Thailand; 4Department of Community Medicine, Faculty of Medicine, Khon Kaen University, Khon Kaen 40002, Thailand; spiyat@kku.ac.th

**Keywords:** body composition, blood lipids, physical health, overweight, Tai Chi

## Abstract

The aim of this study was to evaluate the effects of Bafa Wubu Tai Chi (BW-TC) and traditional He-style Tai Chi (TH-TC) exercises on physical health risk factors in overweight male college students and to compare the effectiveness of the two Tai Chi exercise forms in improving these risk factors. Methods: Eighty-one overweight male university students between the ages of 18 and 23 were randomly assigned in a 1:1:1 ratio to the BW-TC group, TH-TC group, and control group (CG). The Tai Chi exercise training consisted of 12 weeks, three times a week, for 60 min per session. The CG attended three health lectures and maintained their normal study routine. The outcomes were body composition and blood lipids. Data were collected at baseline and post-intervention and analyzed using one-way ANOVA and mixed-design ANOVA. Results: At baseline, there were no significant differences in demographic characteristics and assessed parameters (*p* > 0.05) among the groups. The BW TC and TH TC groups both significantly decreased their body weight (2.69 kg, 2.04 kg, respectively), body mass index (0.90 kg/m^2^, 0.67 kg/m^2^, respectively), body fat percentage (1.46%, 1.10%, respectively), low-density lipoprotein cholesterol (8.82 mg/dL, 9.27 mg/dL, respectively), total cholesterol (8.57 mg/dL, 9.34 mg/dL, respectively) and triglycerides (10.14 mg/dL, 10.63 mg/dL, respectively); and increased their muscle mass (−0.56 kg, −1.13 kg, respectively) and high-density lipoprotein cholesterol (−5.77 mg/dL, −6.37 mg/dL, respectively). Multiple comparisons showed that both Tai Chi groups were significantly better than the CG in improving the evaluated parameters. Conclusions: Two types of Tai Chi interventions were effective in improving body composition and blood lipids in overweight university students, without significant differences between the two.

## 1. Introduction

Being overweight or obese is a significant public health issue with major impacts on various bodily functions [1]. The high incidence of overweight and obese individuals, coupled with a lack of physical activity, increases the risk of chronic diseases such as metabolic syndrome, type 2 diabetes [2] and coronary heart disease [3,4]; ultimately, this can lead to cardiovascular disease [5], psychological health issues [6], musculoskeletal disorders [7], dementia, depression [8] and premature death [9]. These conditions, in turn, have negative impacts on quality of life, work efficiency and healthcare costs. Research has found that 22.7% of males and 8.4% of females among Chinese university students 19–22 years of age are overweight or obese [10]; therefore, overweight and obese Chinese university students represent a major social problem.

Previous research has demonstrated that exercise can improve body composition and lower lipid concentrations [11,12,13], thereby improving lipid distribution [14,15,16], reducing potential risk factors associated with being overweight and enhancing overall health. Tai Chi, an ancient mind–body exercise originating in China [17], combines body movement, breathing and mindfulness in a smooth, continuous manner, enabling practitioners to attain a harmonious state of their body and mind [18]; it has received increasing attention as a low-impact exercise with potential health benefits. While Tai Chi has been shown to improve the lipid profile of overweight individuals [19,20,21,22,23], previous research has primarily focused on the benefits of simplified or modified Tai Chi [24,25,26,27].

Even though Tai Chi has many benefits, research on the health benefits of different styles of Tai Chi for the overweight population, such as Bafa Wubu Tai Chi (BW-TC) and traditional He style Tai Chi (TH-TC) exercises, remains limited, and support for the health benefits of traditional Tai Chi exercise remains weak. Because overweight university students often have difficulty finding effective, enjoyable exercise plans to improve their physical health, comparing the effects of these two types of Tai Chi is particularly meaningful, as they may be feasible choices for this population, even though further research is needed to understand the effects thereof on health risk factors such as blood lipids and body composition. BW-TC exercise is a simplified Tai Chi routine promoted by the General Administration of Sport in China, with simple and easy-to-learn movements consisting of 16 forms, each of which takes approximately three minutes [26]; in contrast, TH-TC exercise is a traditional form of Tai Chi with complex movements consisting of 72 forms, each of which takes approximately 5–8 min to complete [28]. Notably, there is currently no research exploring the effects of BW-TC and TH-TC exercises on overweight and obese individuals.

In this study, we conducted a randomized controlled trial to investigate the effects of two types of Tai Chi exercise training on health risk factors in overweight male university students. The hypothesis of this study was that both BW-TC and TH-TC exercises would have a significant effect on the body composition and blood lipids of overweight university students, with potentially greater advantages being observed with TH-TC exercise. The aim of this study was to evaluate the effects of BW-TC and TH-TC exercises on body composition and blood lipids in overweight university students and to compare how effectively these two types of Tai Chi improved these parameters. The results of this study will help to clarify the potential benefits of Tai Chi exercise for improving the health of overweight university students and provide guidance for future interventions in this population.

## 2. Materials and Methods

### 2.1. Ethical Approval

This study protocol was approved by the Khon Kaen University Ethical Committee for Human Research (no. HE642132). The trial was registered with the Chinese Clinical Trial Registry (registration number ChiCTR2200059427) and carried out in accordance with the Declaration of Helsinki. All participants signed the consent form for the experiment and were informed of the purpose of the study, the procedures involved and the benefits of engaging in this exercise program.

### 2.2. Participants

G*Power (Version 3.1.9.4) was used to estimate the sample size. The *F* test with “ANOVA: Fixed effects, omnibus, one-way” was selected and “a priori: Compute required sample size” was used for the analysis. The effect size convention was set to the maximum value of *f* = 0.40; the α err prob was set to 0.05; the power value (i.e., 1 β err prob) was set to 0.80; and the number of groups was set to 3. The total sample size was 66 people, with 22 people in each group. Assuming a 20% dropout rate, 27 participants were required per group, resulting in a final sample size of 30 subjects per group to improve the test power further. Due to the impact of the COVID-19 pandemic, only male college students from Jiaozuo Normal University, in Jiaozuo City, Henan Province, China, were recruited to be research subjects. In accordance with the recruitment requirements and guidelines, a total of 123 male college students, 18–23 years of age, participated in the qualification screening, which was advertised on posters in the college student activity center and on the sports field, and 90 eligible volunteers were enrolled; however, nine volunteers withdrew midway through the study, leaving a final sample size of 81 subjects.

The inclusion criteria were male college students in their first or second year of university, 18–23 years of age, with a body mass index (BMI, kg/m^2^) between 24 and 27.8 (i.e., per the Chinese overweight standard). Exclusion criteria included regular long-term Tai Chi athletes, sports association members, individuals with severe cardiovascular disease or musculoskeletal system disease and those with a BMI less than 24 or greater than 27.9 (i.e., per the Chinese overweight standard). A flow chart outlining the participants’ entire intervention process and the effects of progress in the trial, which was designed, analyzed and interpreted according to the Consolidated Standards for Reporting Trials (CONSORT) criteria is provided in Figure 1.

### 2.3. Experimental Procedure

This experimental study was a randomized controlled trial. The project manager used web programming at Randomizer.org to randomly divide the participants into three groups: the BW-TC group, the TH-TC group and the control group (CG). The BW-TC and TH-TC groups both received Tai Chi training three times a week for 12 weeks, with each class lasting 60 min and including 15 min of warm-up exercises, 30 min of Tai Chi and 15 min of cool-down exercises; static stretching was used as a warm-up and cool-down component before and after the intervention [29]. Eight static stretching exercises were included, namely (1) neck flexion and neck extension, (2) upper back stretch, (3) forearm stretch, (4) lower back and side stretch, (5) seated low back stretch, (6) hamstring and lower back stretch, (7) hip adductor and lower back stretch and (8) calf stretch. During the warm-up and cool-down phases, each stretching exercise required participants to hold their best stretch position for a minimum of 30 s to achieve the desired stretching effect, while the CG only received health lectures once a month for three months. The intervention period was from 18 October 2021 to 7 January 2022. During the experiment, all three groups maintained their normal diet and daily life and adhered to COVID-19 requirements by wearing masks, sanitizing and maintaining social distancing.

The training sessions were scheduled during the students’ daily activity time on Mondays, Wednesdays and Fridays from 5:30 p.m. to 6:30 p.m. All assessments were conducted on Saturdays and Sundays with 30 participants in each group, and the exercise programs were performed under the direction of the researchers or trained research assistants. Prior to the intervention, all subjects underwent a baseline test that included an evaluation of basic personal physical attributes such as gender, age, height, weight and BMI; body composition and blood lipid measurements were also evaluated. After 12 weeks of Tai Chi training, all 81 subjects were reassessed using the same variables as before the intervention.

### 2.4. Intervention

BW-TC group: BW-TC exercise was used for this intervention. The movements of this form are known for their simplicity, reasonable posture and ease of learning and practice, with each set taking approximately three minutes to complete [26]. Participants practiced Tai Chi continuously for 30 min at an intensity ranging from 50 to 70% of their personal maximum heart rate. Qualified Tai Chi instructors led the participants and provided guidance throughout the practice while emphasizing caution during the exercises.

TH-TC Group: TH-TC exercise was used for this intervention. The book *He Style Taijiquan* written by Youlu He was selected for its instructional content [30]. This type of Tai Chi has 72 movements that use circles and arcs as the form of movement, and the body’s weight of gravity fluctuates continuously [31]; it takes 5–8 min to practice a complete set of movements. Participants also practiced Tai Chi continuously for 30 min at an intensity ranging from 50 to 70% of their personal maximum heart rate. Qualified Tai Chi instructors led the participants and explained the exercises while highlighting reasonable precautions.

Control Group: Instead of receiving an exercise intervention, the CG attended three health lectures, held on the Monday afternoon of the first, fifth and ninth weeks of the experiment (the contents of the lectures were healthy diet concepts, mental health cultivation and knowledge presentation on physical activity to promote health, respectively), while maintaining their normal daily routines.

### 2.5. Measurement Procedure

#### 2.5.1. Baseline Measurement

Baseline: The height and weight of the participants were measured at baseline using a Bingyu RGZ 160 height–weight scale (Changzhou, China). These measurements were taken while each participant stood on an unsupported spring scale with their weight evenly distributed on their feet and heels, their buttocks and upper back touching the rangefinder scale, and their head not touching the scale. The height and weight measurements were recorded in centimeters and kilograms, respectively, and rounded to one decimal place. BMI was calculated by dividing each participant’s weight by the square of height, accurate to one decimal place [32]. Age data were obtained through the personal characteristics table.

#### 2.5.2. Outcome Measures

Body composition: Body composition was assessed by measuring body weight, BMI, body fat percentage (FP, %) and muscle mass (MM, %) using a BCA 2A Body Composition Analyzer (Tongfang Health Technology Company, Beijing, China). The participants were tested on an empty stomach on Saturday and Sunday mornings in accordance with the test guidelines, with instructions to avoid strenuous exercise the day prior to testing. During their measurements, each participants stood barefoot on the base of the apparatus with their heels on the ground electrodes, pressed the soles of their feet against the oval electrodes in front and held the hand electrodes with both hands, holding for 40–60 s until all measurements were completed [33].

Blood lipid: The blood-lipid parameters assessed were high-density lipoprotein cholesterol (HDL-C, mg/dL), low-density lipoprotein cholesterol (LDL-C, mg/dL), total cholesterol (TC, mg/dL) and triglycerides (TG, mg/dL); an AU5821 Clinical Chemistry Analyzer (Beckman Coulter Diagnostics, Brea, CA, USA) was used to analyze the blood lipids. Due to the specialized nature of the blood lipid measurement, a qualified nurse completed the measurements and evaluations. The participants provided a 2 mL fasting blood sample before 9:00 a.m. on Saturday and Sunday before and after the intervention. They were instructed to avoid alcohol and fast for 12 h before the test, and they received psychological counselling throughout the study. Lipid data were converted from mmol/L to mg/dl and the cholesterol (i.e., TC, LDL-C, HDL-C) and TG data were multiplied by 38.67 and 88.57, respectively [21].

### 2.6. Statistical Analysis

Data were collected using Excel 2019, and a statistical analysis was performed using IBM^®^ SPSS^®^ Statistics Version 26.0 (IBM Corporation, Armonk, NY, USA). The data were presented as mean ± standard deviation (SD), and a 95% confidence interval (CI) was adopted. Descriptive statistics were used to summarize the participants’ baseline characteristics and one-way ANOVA tests were used to compare group differences at baseline. Following the 12-week intervention, a mixed-design ANOVA was used to examine the effects of the intervention and control groups on the dependent variables (i.e., body composition and blood lipids), with a within factor of time effects (i.e., before and after) and a between factor of time-by-group effects (i.e., BW-TCG, TH-TCG and CG). The least significant difference was used to compare changes within each group before and after the intervention and differences between the groups. A dependent *t*-test was performed to determine changes within each group, and statistical significance was set at *p* < 0.05.

## 3. Results

A total of 81 overweight male college students with a BMI between 24 and 27.9 kg/m^2^, who were 18–23 years of age, participated in this study. There were no statistically significant differences among the three groups in the demographic variables (i.e., mean age, weight, height and BMI), and no statistically significant differences were observed among the groups at baseline (see Table 1).

This study examined the influencing factors and trends of the evaluated variables in 81 overweight male college students across three groups and two points in time. The results indicated that the interactions between the time effects and the time-by-group effects of evaluation variable parameters were significant (*p* < 0.05) (see Table 2). The researchers subsequently conducted a detailed analysis of the simple time effect and the simple time-by-group effect to compare the differences of each group before and after the intervention, as well as the differences among the three groups after the 12-week intervention.

After 12 weeks of intervention, the body weight, BMI, body fat percentage, LDL-C, TC and TG of the BW-TC and TH-TC groups significantly decreased from the zero-week baseline measurement to the 12-week follow-up (*p* < 0.001). Furthermore, the muscle mass and HDL-C of these groups significantly increased (*p* < 0.001). Notably, no statistically significant differences were observed in the CG (see Table 3).

In multiple comparisons, the BW-TC group demonstrated a significant decrease in body weight (*p* = 0.031), BMI (*p* = 0.001), body fat percentage (*p* = 0.005), LDL-C (*p* = 0.001), TC (*p* = 0.042) and TG (*p* = 0.001), along with a significant increase in HDL-C (*p* = 0.047) compared to the CG. The TH-TC group also showed a significant decrease in body fat percentage (*p* = 0.001), LDL-C (*p* = 0.001), TC (*p* = 0.034) and TG (*p* = 0.001), in addition to a significant increase in muscle mass (*p* = 0.034) and HDL-C (*p* = 0.025) compared to the CG. No statistically significant differences were observed in the body composition and blood lipids of the TH-TC and BW-TC groups (see Table 4).

## 4. Discussion

### 4.1. Changes in Body Composition

Tai Chi is a physical activity that can lead to changes in body composition due to increased energy expenditure. Previous studies have shown significant improvements in weight, BMI, body fat percentage and muscle mass in different populations after a 12-week Tai Chi intervention [34,35,36]; this was also observed in our study. The BW-TC and TH-TC groups both demonstrated significant improvements in body composition, while no significant changes were observed in the CG (Table 3).

During the Tai Chi practice sessions, the subjects were instructed to maintain their abdominal breathing, meditation and physical and mental relaxation; deep breathing and meditative states contribute to relaxation and stress reduction in the mind and body and may have a positive effect on body composition by modulating hormonal and inflammatory processes [37]. The decreased body weight and BMI documented in our study may be attributed to these factors. The decreased body fat percentages were possibly due to the design of our Tai Chi training program and the exercise participants. Our Tai Chi training program involved continuous practice for 30 min, with exercise intensity at 50–70% of the participants’ maximum heart rate; aerobic exercise mainly consumes fat to provide energy [38], which explains our finding that body fat percentages decreased in the BW-TC and TH-TC groups (Table 3). During Tai Chi practice, the body’s center of gravity repeatedly moves up and down, which is the equivalent of a resistance exercise; these movements stimulate the leg muscles, which increases muscle strength [39]. Muscle strength correlates with muscle mass [40,41,42] and may be a key factor in the increased muscle mass observed in the study subjects.

### 4.2. Changes in Blood Lipids

Exercise is a key factor in regulating fatty acid oxidation [43] and light- and moderate-intensity training will further increase fatty acid oxidation [44]. As expected in our study, there were statistically significant differences between baseline and post-intervention mean lipid variables in the BW-TC group after the 12-week Tai Chi training program, i.e., HDL-C increased by 12% to 5.769 mg/dL, LDL-C increased by 9% to 8.22 mg/dL, TC increased by 7% to 8.57 mg/dL and TG decreased by 10% to 10.14 mg/dL. Significant statistical differences were also observed between baseline and post-intervention mean lipid variables in the TH-TC group, i.e., HDL-C increased by 13% to 6.37 mg/dL, LDL-C decreased by 10% to 9.27 mg/dL, TC decreased by 6% to 9.34 mg/dL and TG decreased by 10% to 10.63 mg/dL (Table 3). Even though our study consisted of overweight male college students, these findings broaden the audience for the effects of Tai Chi training on blood-lipid levels. The CG, which did not participate in Tai Chi training, did not report significant improvements in their blood-lipid assessment variables after 12 weeks compared to their baseline (Table 3); we asked the CG to maintain a daily diet and to study throughout the experiment, and we provided three healthcare sessions, which may be why there were no significant changes in the CG blood-lipid variables. Notably, even though our pilot study showed the TH-TC exercise intensity was slightly higher than that of BW-TC, the different types of Tai Chi yielded no significant difference in the effect of blood lipids on overweight male college students (Table 4), so future studies may require a longer length of time to discover these differences.

The findings of our study align with those of earlier studies conducted by Shou et al. (2019), Shih-Chueh et al. (2010) and Shi et al. (2014) [19,20,21,22,23,45,46]. Tai Chi is a combination of aerobic exercise and resistance exercise; during exercise, skeletal muscle deconstructs adipose tissue and oxidizes fatty acids, which mobilizes lipids throughout the body, leading to control tissue uptake of fatty acids, intracellular lipid transport and mitochondrial β oxidation and increased systemic fat oxidation, thereby lowering blood lipid levels [47,48]. This is the probable physiological mechanism by which Tai Chi training improves blood lipids through a combination of aerobic and resistance exercise.

The first hypothesis of this study was that both BW-TC and TH-TC exercises would have a significant effect on the body composition and blood lipids of overweight university students, and our research findings confirmed this hypothesis. However, the second hypothesis that TH-TC exercise would have a more significant advantage than BW-TC exercise in improving body composition and blood lipids was not supported. The results showed no significant differences (*p* > 0.05) between the two types of Tai Chi in terms of body weight (kg), BMI (kg/m^2^), FP (%), MM (%), HDL-C (mg/dL), LDL-C (mg/dL), TC (mg/dL) and TG (mg/dL) (Table 4). Despite the differences in exercise methods, styles, number of movements and requirements between the two types of Tai Chi, it may require a longer intervention period in the future to reveal the differences between them.

### 4.3. Limitations and Future Studies

While this study confirmed the benefits of Tai Chi training, there are several limitations that should be considered. First, the study did not evaluate the impact of Tai Chi on mental health factors, such as anxiety and depression. Second, because the study only included overweight male students, it is unclear whether the findings can be extrapolated to normal or obese populations or to females. As such, future research should investigate the effects of these Tai Chi programs on mental health and examine a more diverse population. Moreover, longer training periods and further additions to the research are needed to fully comprehend the different effects of the two Tai Chi programs on overweight and obese college students.

### 4.4. Innovative Contribution

This study evaluated the effects of two different types of Tai Chi on body health risk factors in overweight university students; the results showed that both types of Tai Chi effectively improved body composition and blood lipid levels, with no significant difference between them. This provides new evidence for the application of Tai Chi in overweight populations and sheds light on its physiological mechanisms related to fatty acid oxidation and systemic lipid regulation. Therefore, this study is innovative in comprehensively assessing the impacts on body health, comparing different forms of Tai Chi, and revealing physiological mechanisms.

## 5. Conclusions

The present study determined that the BW-TC and TH-TC exercise forms of Tai Chi can both effectively improve the body composition and lower the blood lipids of overweight students, with neither showing superiority over the other. Accordingly, both Tai Chi programs can serve as viable exercise alternatives to mitigate the risk of chronic diseases among overweight and obese college students.

## Figures and Tables

**Figure 1 ijerph-20-06323-f001:**
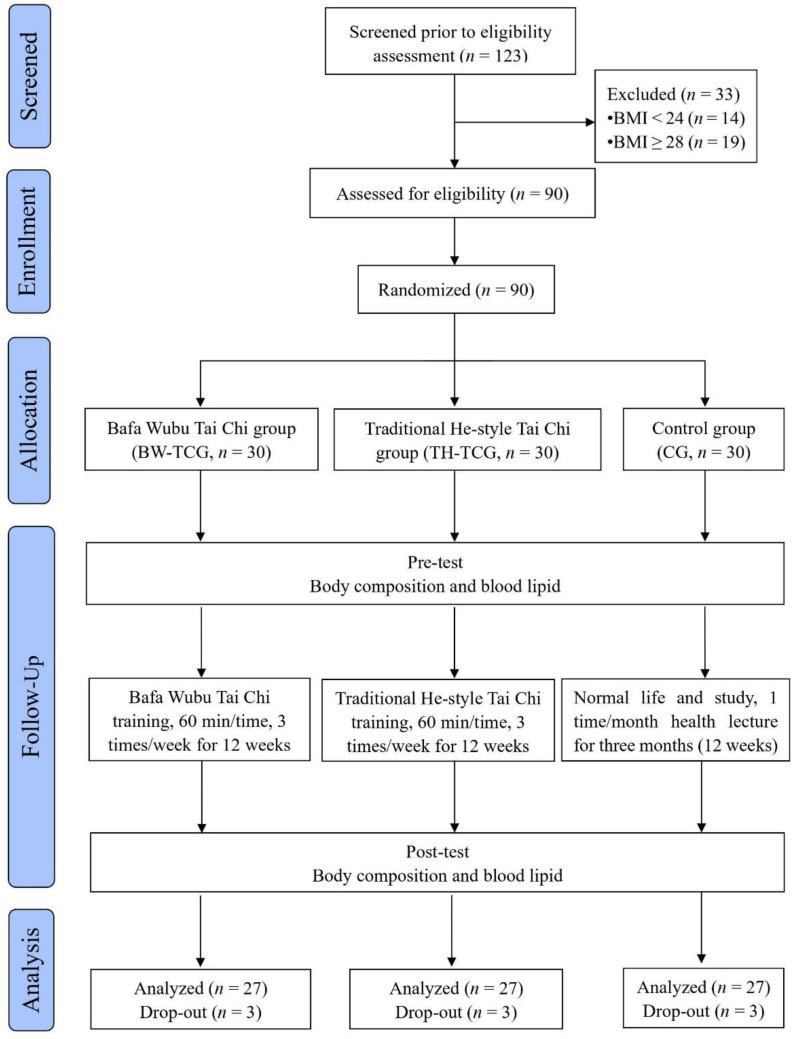
CONSORT flow diagram.

**Table 1 ijerph-20-06323-t001:** Demographic, baseline body composition and blood lipids of study participants.

Outcomes	BWTCG (*n* = 27)	THTCG (*n* = 27)	CG (*n* = 27)	*p*-Value
	(mean ± SD)	(mean ± SD)	(mean ± SD)	
Age (year)	18.48 ± 0.64	18.41 ± 0.75	18.67 ± 0.62	0.350
Height (cm)	175.56 ± 5.19	174.41 ± 5.28	175.41 ± 4.14	0.646
Body weight (kg)	80.33 ± 5.81	80.50 ± 5.33	80.66 ± 4.29	0.974
BMI (kg/m^2^)	26.07 ± 1.30	26.45 ± 1.12	26.23 ± 1.12	0.496
FP (%)	27.87 ± 3.70	27.08 ± 3.30	28.71 ± 2.80	0.195
MM (%)	32.56 ± 2.31	32.60 ± 2.54	32.26 ± 2.36	0.852
HDL-C (mg/dL)	49.17 ± 8.0	49.23 ± 7.97	49.44 ± 9.72	0.993
LDL-C (mg/dL)	87.75 ± 10.80	88.43 ± 10.70	87.98 ± 10.72	0.973
TC (mg/dL)	152.97 ± 15.74	153.39 ± 13.83	152.45 ± 16.02	0.974
TG (mg/dL)	100.77 ± 12.39	101.56 ± 12.29	101.04 ± 12.31	0.972

Note: One-way ANOVA test; BWTCG, Bafa Wubu Tai Chi group; THTCG, traditional He-style Tai Chi group; CG, control group; BMI, body mass index; FP, fat percentage; MM, muscle mass; HDL-C, high-density lipoprotein cholesterol; LDL-C, low-density lipoprotein cholesterol; TC, total cholesterol; TG, triglycerides; cm, centimeter; kg, kilogram; m, meter.

**Table 2 ijerph-20-06323-t002:** Baseline and after intervention mean and standard deviation, F statistic for time effects and time-by-group effects, *p*-values and partial eta squared of parameters for the three groups.

Outcomes	Group	Mean ± SD		Time Effects	*p*-Value	PES	Time × Group Effects	*p*-Value	PES
		Pre	Post						
	BWTCG	80.33 ± 5.81	77.65 ± 5.88	F (1, 78) = 70.984			F (2, 78) = 25.776		
Body weight (kg)	THTCG	80.50 ± 5.33	78.45 ± 5.28		0.001	0.476		0.001	0.398
	CG	80.66 ± 4.30	80.93 ± 5.31						
	BWTCG	26.07 ± 1.30	25.17 ± 1.19	F (1, 78) = 66.481			F (2, 78) = 24.293		
BMI (kg/m^2^)	THTCG	26.45 ± 1.12	25.78 ± 1.10		0.001	0.460		0.001	0.384
	CG	26.23 ± 1.12	26.31 ± 1.31						
	BWTCG	27.87 ± 3.70	26.41 ± 3.24	F (1, 78) = 75.588			F (2, 78) = 29.092		
FP (%)	THTCG	27.08 ± 3.30	25.97 ± 3.38		0.001	0.492		0.001	0.427
	CG	28.71 ± 2.80	28.88 ± 2.66						
	BWTCG	32.56 ± 2.31	33.11 ± 2.27	F (1, 78) = 40.791			F (2, 78) = 10.188		
MM (%)	THTCG	32.60 ± 2.54	33.73 ± 2.40		0.001	0.343		0.001	0.207
	CG	32.26 ± 2.36	32.36 ± 2.32						
	BWTCG	49.17 ± 8.07	54.94 ± 8.71	F (1, 78) = 288.186			F (2, 78) = 54.363		
HDL-C (mg/dL)	THTCG	49.23 ± 7.97	55.59 ± 7.93		0.001	0.787		0.001	0.582
	CG	49.44 ± 9.72	50.01 ± 10.12						
	BWTCG	87.75 ± 10.80	78.93 ± 8.40	F (1, 78) = 157.766			F (2, 78) = 33.061		
LDL-C (mg/dL)	THTCG	88.43 ± 10.70	79.16 ± 8.65		0.001	0.669		0.001	0.459
	CG	87.98 ± 10.72	87.45 ± 10.44						
	BWTCG	152.97 ± 15.74	144.39 ± 14.65	F (1, 78) = 335.698			F (2, 78) = 92.558		
TC (mg/dL)	THTCG	153.39 ± 13.83	144.05 ± 13.44		0.001	0.811		0.001	0.704
	CG	152.45 ± 16.02	152.72 ± 16.16						
	BWTCG	100.77 ± 12.39	90.64 ± 9.65	F (1, 78) = 159.763			F (2, 78) = 33.326		
TG (mg/dL)	THTCG	101.56 ± 12.29	90.93 ± 9.92		0.001	0.672		0.001	0.461
	CG	101.04 ± 12.31	100.41 ± 11.87						

Note: Mixed design ANOVA test; adjustment for multiple comparisons, least significant difference. BWTCG, Bafa Wubu Tai Chi group; THTCG, traditional He-style Tai Chi group; CG, control group; PES, partial Eta squared; BMI, body mass index; FP, fat percentage; MM, muscle mass; HDL-C, high-density lipoprotein cholesterol; LDL-C, low-density lipoprotein cholesterol; TC, total cholesterol; TG, triglycerides; kg, kilogram; m, meter.

**Table 3 ijerph-20-06323-t003:** Mean differences, *p*-value and 95% confidence interval for differences in the paired t-test, before and after 12 weeks of intervention within each group.

Outcomes	BWTCG	*p*-Value	THTCG	*p*-Value	CG	*p*-Value
	Md (95% CI)		Md (95% CI)		Md (95% CI)	
Body weight (kg)	2.69 (2.08 to 3.29)	0.001	2.04 (1.44 to 2.65)	0.001	−0.27 (−0.88 to 0.34)	0.386
BMI (kg/m^2^)	0.90 (0.69 to 1.11)	0.000	0.67 (0.46 to 0.88)	0.000	−0.09 (−0.30 to 0.12)	0.402
FP (%)	1.46 (1.15 to 1.78)	0.001	1.10 (0.79 to 1.42)	0.001	−0.17 (−0.49 to 0.15)	0.288
MM (%)	−0.56 (−0.88 to −0.24)	0.001	−1.13 (−1.45 to −0.81)	0.001	−0.10 (−0.42 to 0.22)	0.536
HDL-C (mg/dL)	−5.77 (−6.63 to −4.91)	0.001	−6.37 (−7.23 to −5.51)	0.001	−0.57 (−1.43 to 0.29)	0.189
LDL-C (mg/dL)	8.82 (7.12 to 10.53)	0.001	9.27 (7.56 to 10.97)	0.001	0.53 (−1.17 to 2.23)	0.538
TC (mg/dL)	8.57 (7.47 to 9.68)	0.001	9.34 (8.23 to 10.44)	0.001	−0.28 (−1.39 to 0.83)	0.617
TG (mg/dL)	10.14 (8.19 to 12.08)	0.001	10.63 (8.68 to 12.57)	0.001	0.62 (−1.322 to 2.57)	0.525

Note: paired *t*-test. BWTCG, Bafa Wubu Tai Chi group; THTCG, traditional He-style Tai Chi group; CG, control group; CI, confidence Interval; Md, mean difference; BMI, body mass index; FP, fat percentage; MM, muscle mass; HDL-C, high-density lipoprotein cholesterol; LDL-C, low-density lipoprotein cholesterol; TC, total cholesterol; TG, triglycerides; kg, kilogram; m, meter.

**Table 4 ijerph-20-06323-t004:** Mean differences, *p*-value and 95% confidence interval for differences for pairwise comparison after 12 weeks intervention among the three groups.

Outcomes	BWTCG-CG	*p*-Value	THTCG-CG	*p*-Value	THTCG-BWTCG	*p*-Value
	Md (95% CI)		Md (95% CI)		Md (95% CI)	
Body weight (kg)	−3.28 (−6.26 to −0.30)	0.031	−2.47 (−5.45 to 0.51)	0.102	0.80 (−2.18 to 3.78)	0.593
BMI (kg/m^2^)	−1.15 (−1.80 to 0.50)	0.001	−0.54 (−1.19 to 0.11)	0.104	0.61 (−0.04 to 1.26)	0.065
FP (%)	−2.47 (−4.16 to −0.79)	0.005	−2.91 (4.59 to −1.22)	0.001	−0.43 (2.12 to 1.25)	0.610
MM (%)	0.75 (−0.51 to 2.02)	0.240	1.37 (0.10 to 2.63)	0.034	0.62 (−0.65 to 1.88)	0.336
HDL-C (mg/dL)	4.92 (0.07 to 9.78)	0.047	5.58 (0.72 to 10.44)	0.025	0.66 (−4.20 to 5.52)	0.788
LDL-C (mg/dL)	−8.52 (−13.51 to −3.53)	0.001	−8.29 (−13.28 to −3.30)	0.001	0.23 (−4.76 to 5.22)	0.927
TC (mg/dL)	−8.33 (−16.35 to −0.32)	0.042	−8.67 (−16.69 to −0.66)	0.034	−0.34 (8.36 to 7.67)	0.932
TG (mg/dL)	−9.78 (−15.48 to −4.07)	0.001	−9.48 (−15.18 to −3.78)	0.001	0.30 (−5.41 to 6.00)	0.918

Note: Mixed design ANOVA test; adjustment for multiple comparisons, least significant difference. BWTCG, Bafa Wubu Tai Chi group; THTCG, traditional He-style Tai Chi group; CG, control group; CI, confidence Interval; Md, mean difference; BMI, body mass index; FP, fat percentage; MM, muscle mass; HDL-C, high-density lipoprotein cholesterol; LDL-C, low-density lipoprotein cholesterol; TC, total cholesterol; TG, triglycerides; kg, kilogram; m, meter.

## Data Availability

Not applicable.

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
