# Peer review of "Comparing the Effects of Bafa Wubu Tai Chi and Traditional He-Style Tai Chi Exercises on Physical Health Risk Factors in Overweight Male College Students: A Randomized Controlled Trial"

_ijerph, 2023, doi:10.3390/ijerph20146323_

Round 1
Reviewer 1 Report
Niu_Tai-Chi_IJERPH_2023
Reviewer report
Thank you for presenting your manuscript. The work was carefully executed.
The main question addressed by the research is clear and well structured, that is, to evaluate the effects of BW TC and TH TC on body composition and blood lipids in overweight university students and compare how effectively these two types of Tai Chi improved these parameters. The topic is original and relevant in the field because no previous studies have compared the effects between BW TC and TH TC. The study is well designed, statistical analysis are complex and analyze deeply the relations between the implied factors time and group.
The conclusions are consistent and are based only in the outcomes analyzed and the arguments of the discussion that address the main question of the study. The references are appropriate. The tables and figures are of enough quality and quantity.
I have some minor concerns about improvements regarding the methodology and the discussion highlighted bellow.
Specific comments
2. Materials and Methods
Figure 1. CONSORT flow diagram. The sentence 3 times health for 12 weeks, referring to Follow up of the control group is not very clear. I think it is better to say “1 time/month health lecture for three months (12 weeks)”.
Line 144. Please, can you add a sentence explaining the content of the three health lectures?
Line 175: Why do you “multiplied by 38.67 and 88.57, respectively”? Please explain.
Discussion
Please add one paragraph at the end of the discussion section highlighting the innovations of the study to the subject area.
Reviewer 2 Report
The authors must be commended for carrying out a study regarding the influence of two ancient exercise types on body composition and lipid profile. This topic is interesting, the research methodology used in the study is appropriate, and the manuscript is written with good clarity. However, some issues need to be taken into consideration.
Abstract
Line 15: ‘’Ninety overweight male university students...’’. You stated in the ‘’participants’’ section (methods) that 81 subjects were the final sample size. Please correct.
Add ‘’respectively’’ after each par of indicators in the brackets: ‘’The BW TC and TH TC groups both significantly decreased their body weight (2.69 kg, 2.04 kg, respectively)...’’
Introduction
Please add a hypothesis.
Methods
Line 115 and 116: Please elaborate on warm-up and cool-down exercise.
I do not quite understand the ‘’outcome measures’’ part. First of all, I do not understand the ‘’outcome measures’’ term. You are referring to the parameters used in the study, it is unusual to use the ‘’outcome measures’’ term. Secondly, I do not understand the following classification; baseline measurement-outcome measures. You measured the same parameters at the baseline and the end of the experimental intervention. I suggest renaming this whole section to - ‘’measurement procedure’’.
I suggest naming all measured parameters along with the abbreviations and the units for each parameter.
I suggest adding/replacing the parameter MM in kg with MM in % (proportion of muscle mass according to body mass). It is more representative for observing the proportion of muscle mass.
Results
Sometimes you use an exact p value (p=0.031 e.g.) and sometimes you don’t (p<0.001 e.g.). Please unify.
Discussion
Line 256: ‘’ Our Tai Chi 254 training program involved continuous practice for 30 minutes, with exercise intensity at 50–70% of the participants’ maximum heart rate...’’. You didn’t mention this in the methods (intervention) part. Please add this information.
Line 261: ‘’ Muscle 260 strength correlates with muscle mass [39]...’’. Please add some new reference.
After each important statement about observed results, please add information about the place in the results section where we can see mentioned results (Table 1 e.g.).
Please elaborate on the non-existence of the differences between the two Tai Chi programs.
Round 2
Reviewer 2 Report
Dear Authors,
Thank you for taking into consideration my comments and suggestions. From my pont of view, manuscripy is now suitable for publishing.
Best regards.